# High Specific Capacitance of the Electrodeposited MnO_2_ on Porous Foam Nickel Soaked in Alcohol and its Dependence on Precursor Concentration

**DOI:** 10.3390/ma13010181

**Published:** 2020-01-01

**Authors:** Ming Zhang, Xiaoli Dai, Cuixian Zhang, Yuanwu Fuan, Dingyu Yang, Jitao Li

**Affiliations:** 1College of Optoelectronic Technology, Chengdu University of Information Technology, Chengdu 610225, China; a981671793@163.com (M.Z.); Xiaoli_DaiZH@163.com (X.D.); ZCX1306@outlook.com (C.Z.); fuanyuanwu@163.com (Y.F.); 2School of Precision Instruments and Optoelectronics Engineering, Tianjin University, Tianjin 300072, China

**Keywords:** MnO_2_, precursor concentration, electrodeposition, supercapacitor, high specific capacitance

## Abstract

In this work, we used the mixed solution of manganese acetate and sodium sulfate to deposit manganese dioxide on the three-dimensional porous nickel foam that was previously soaked in alcohol, and then the effects of solution concentrations on their capacitance properties were investigated. The surface morphology, microstructure, elemental valence and other information of the material were observed by scanning electron microscope (SEM), Transmission Electron Microscope (TEM), X-ray photoelectron spectroscopy (XPS), etc. The electrochemical properties of the material were tested by Galvanostatic charge-discharge (GCD), Cyclic Voltammetry (CV), Chronoamperometry (CA), Electrochemical impedance spectroscopy (EIS), etc. The MnO_2_ electrode prepared at lower concentrations can respectively reach a specific capacitance of 529.5 F g^−1^ and 237.3 F g^−1^ at the current density of 1 A g^−1^ and 10 A g^−1^, and after 2000 cycles, the capacity retention rate was still 79.8% of the initial capacitance, and the energy density can even reach 59.4 Wh Kg^−1^, while at the same time, it also has a lower electrochemical impedance (Rs = 1.18 Ω, Rct = 0.84 Ω).

## 1. Introduction

The emergence of the fossil energy crisis has forced us to use new energy sources such as wind and solar energy. However, these energy sources are usually unstable, and it is very important to store energy with energy storage equipment [1]. The supercapacitor has attracted a great deal of interest due to its high power density, fast charge and discharge, and excellent cycle performance characteristics [2,3]. MnO_2_ is rich in content, is cheap, green and non-polluting, and has a high theoretical specific capacity (1370 F g^−1^) and a wide potential window, which is considered to be one of the most attractive electrode materials [4,5]. However, it is often difficult for MnO_2_ to maintain a large specific capacitance at a high charge and discharge rate due to its low conductivity [6,7]. Synthesizing manganese dioxide with different morphologies to obtain a large specific surface area by different methods is an effective means to improve the specific capacitance of manganese dioxide [8,9,10]. Nanoflowers [11], nanorods [12], nanospheres [8], nanowires [13] and nanotubes [14] are currently reported. The electrodeposition method has the characteristics of being fast, simple, pollution-free and easy to control, and the prepared material often have higher specific capacitance [15]. 

In addition, conventional electrode preparation methods require the incorporation of a binder and an active material on the collector, which is detrimental to the improvement in electrochemical performance [16,17]. The three-dimensional porous nickel foam used as the substrate for electrodeposition has a higher active material loading mass compared to the conventional planar substrate, and the three-dimensional structure of the nickel foam substrate can also shorten the ion transmission distance and provide a highly conductive network. For the electrodeposition method, the main current research focuses on different deposition methods and different voltages and currents [16,17,18], and in fact, the concentration of precursors also has a great effect on electrodeposition, as it is necessary to study the effect of precursor concentration on electrodeposition.

In this work, we improved electrodeposition process and systematically studied the surface morphology, microstructure and electrochemical properties of manganese dioxide thin films deposited in different precursor concentrations by a constant voltage method. A high-performance manganese dioxide supercapacitor was prepared at a concentration of 0.06 M by immersing alcohol before electrodeposition to make the manganese dioxide film deposition more uniform. The MnO_2_ electrode prepared in the precursor with a concentration of 0.06 M has lower charge transfer resistance (0.84 Ω), and due to the low resistance, this electrode also has high rate performance (237.3F g^−1^ at a current density of 10 A g^−1^) and high energy density (59.4 Wh Kg^−1^, 1 A g^−1^).

## 2. Experimental

### 2.1. Synthesis of MnO_2_

Ni foam (110 PPI, 350 g m^−2^, 1.0 mm thick, 1 × 3.5 cm^−2^) was employed as the substrate for MnO_2_ electrodeposition. A mixed aqueous solution of Mn(CH_3_COO)_2_ and Na_2_SO_4_ tetrahydrate with a concentration of 0.06 M, 0.07 M, 0.08 M and 0.09 M, respectively, were used as a deposition precursors. Firstly, the Ni foam was ultrasonically cleaned with hydrochloric acid, alcohol and deionized water for 5 min for the removal of possible oxides on the surface. Then the cleaned Ni foams were immersed in alcohol for 1 min at room temperature to obtain a more uniform MnO_2_ film during electrodeposition, and where Figure 1a,b shows the different SEM images, it is easy to see that Figure 1b grows more uniformly than Figure 1a. In addition, we compare both with and without alcohol at the same charge by using the chronocoulometry technique (Figure 1c), and found that the electrochemical performance of the sample soaked in alcohol is better. Before the experiment, the voltage of the linear part was selected as the experimental deposition voltage according to the linear sweep voltammetry (LSV) curve of MnO_2_ electrodeposition (Figure 1d). A mixed solution of manganese acetate and sodium sulfate with concentrations of 0.06 M, 0.07 M, 0.08 M and 0.09 M as a precursor were deposited at 0.6 V for 50 s in three-electrode system (Ag/AgCl as the reference electrode and platinum plate electrode as the counter electrode), respectively. Subsequently, all the samples were washed with deionized water and dried at 60 °C in a blower box for 12 h. The mass change Δm before and after deposition was measured by a high precision electronic balance.

### 2.2. Characterization

The morphologies and structures of the MnO_2_ films were characterized by emission scanning electron microscopy (ESEM) (Zeiss ULTRA 55 SEM, 20 KV, Heidenheim, Germany) and transmission electron microscope (TEM, FEI Tecnai G2 F20 with an accelerating voltage of 200 kV, Hillsboro, OR, USA). TEM samples were prepared by scraping the MnO_2_ films out of the Ni foam matrix, dispersing them to alcohol by ultrasonic, dropping them on copper grids and finally drying them at room temperature. The composition of the MnO_2_ films was determined by X-ray photoelectron spectroscope (XPS, Thermo Fisher K-Alpha 250xi, Waltham, MA, USA). The quality of MnO_2_ films were weighed by a high-precision electronic balance (Toledo, XS205DU, d = 0.01 mg, Zurich, Switzerland), weighed 0.2 mg, 0.3 mg, 0.5 mg, 0.6 mg of MnO_2_ (0.06 M, 0.07 M, 0.08 M, 0.09 M), respectively. The electrochemical performance of the MnO_2_ films was tested at room temperature in electrolyte of 1 M Na_2_SO_4_ by CS350 (Wuhan CorrTest Instrument Co. Ltd., Wuhan, China) electrochemical workstation with the MnO_2_/Ni as the working electrode, a platinum plate as the counter electrode, and Ag/AgCl as the reference electrode. The specific capacitance, energy density (*E*) (Wh kg^−1^) and the power density (*P*) (W kg^−1^) of the MnO_2_ at different current densities can be calculated from the constant current charge and discharge curves according to following three Equations [19].
(1)$$C=\frac{I∆t}{∆Vm}$$
(2)$$E=\frac{1}{2}\times \frac{C{\left(∆V\right)}^{2}}{3.6}$$
(3)$$P=\frac{E}{∆t}\times 3600$$

In which *I* is the discharge current (*A*), ∆t is the discharge time, and m represents the mass of active materials (g), ∆V is *t* the potential window (V vs. Ag/AgCl). 

The area ratio capacitance of MnO_2_ can be calculated by the CV curve according to Equation (4), where *v* is the scan rate (V s^−1^), and S is the area of the active materials in electrode (2 cm^−2^).
(4)Cs=(∫IdV)/(v×V×S)

## 3. Results and Discussion

### 3.1. Effect of Alcohol on Deposition of MnO_2_

On the one hand, in 2006, Meng and Yu prepared the Ni(OH)_2_ supercapacitor electrode material in the ethanol and water system, and reported that the proper amount of alcohol could reduce the surface tension between the electrode and the electrolyte liquid boundary to improve the adhesion strength of the precipitate [20]. Similarly, the mentioned reason may apply to this work, in other words, alcohol may improve the surface tension of the Ni foam, which helps MnO_2_ to be better deposited onto the Ni foam. On the other hand, previous reports showed that the Ni particle could catalyze the ectrooxidation of ethanol to produce intermediates such as electrons and protons [21,22]. Further, in 2019, Abolfath Eshghi et al. proposed that there were the interactions between MnO_2_ and ethanol in the electrochemical process, and found that MnO_2_ could catalyze the ectrooxidation of ethanol, too [21]. However, the clear explanation about interactions between MnO_2_ and ethanol was not given in detail. In this work, we guess that the both Ni and MnO_2_ improved alcohol ectrooxidation, and some intermediates such as electrons and protons might take part in the electrodeposited process of MnO_2_, so that the deposition quality of MnO_2_ was improved. (Please note: In this work, we investigate mainly the precursor concentration effects on the electrochemical performances of MnO_2_ deposited on alcohol-soaked Ni foam, and we do not make the further discussion on the alcohol effect mechanism. We hope readers to pay attention to our further work that the alcohol effect mechanism as a unique issue will be studied in detail).

### 3.2. Surface Topography and Microstructure Analysis

The surface topography of the material has a great impact upon performance. Figure 2 shows SEM images of MnO_2_ films deposited with different precursor concentrations, and the internal illustration is a 5 µm scale SEM im(age. All of the manganese dioxide films are uniformly grown on the three-dimensional porous foam nickel stent, and the films are loose, porous structures, a large part of which is attributed to the bubbles generated during the redox process. As shown in Figure 2a–d, the surface of the film is made up of a plurality of nanobars connected to each other, and a large number of nanostrips form a three-dimensional, porous structure. This structure has a larger specific surface area, enabling the active material to sufficiently react with ions in the solution, and thus shortening the electron transport distance. As the concentration of the deposition precursor increases, there is no significant difference in the microscopic morphology of the material surface, but the mass of the active material deposited on the substrate increases, and the mass specific capacity decreases (the electrochemical performance chart behind is easy to see). We conclude that the change in the solubility of the electrodeposition solution in a small range will only cause a change in the deposition quality (film thickness), and while the electrochemical reaction mainly occurs on the surface of the film, lowering the effective utilization of thicker films, the specific capacitance also decreases [23]. However, at lower precursor concentrations, the loading mass of the manganese dioxide film is very low, and the overall capacitance performance is reduced. 

In order to further understand the microstructure of the prepared manganese dioxide, we used ultrasonic waves to shake the manganese dioxide powder from the foamed nickel substrate and observed it (0.06 M, all samples showed a similar structure, so the 0.06 M concentration sample was selected as the representative) by TEM. Figure 3a–d show the TEM images of MnO_2_ nanostrips at different magnifications. The image shows that the growth of manganese dioxide is very uniform. Nanostrips are interlaced and interconnected, this structure is in favor of ion transport and electron transfer, thereby exhibiting good electrochemical performance. HR-TEM images show that the diameter of the MnO_2_ nanostrip is about 30–50 nm, and the length is about 300–500 nm. The spacing between the planes is 0.223 nm (002 plane) and 0.259 nm (301 plane), respectively, which belong to orthorhombic system. 

In order to study the elemental of manganese dioxide and further prove its existence, we have studied it by X-ray photoelectron spectroscopy (XPS). As shown in Figure 4a, full-peak scans demonstrate the presence of Mn and O elements. The 2p orbital of Mn has two distinct peaks, which are Mn2p_3/2_ (642.19 eV) and Mn2p_1/2_ (653.97 eV), while the separation value is 11.78 eV, and this result is in great agreement with that in the work by Therese and Kamath [15], which suggests that the element Mn is in the +4 valence state. The 3s orbital separation energy of Mn is 5.05 eV, indicating that the Mn element is mainly in the +3 valence state and the +4 valence state [24,25,26]. In the energy spectrum of O1s, the peak is mainly located at 529.8 eV (Mn–O–Mn), 531.2 eV (Mn–O–H), and 532.8 eV (H–O–H), and this result is also in excellent agreement with that in the work by Yan et al. [27]. The presence of Mn–O–H and H–O–H reveals the hydrous nature [28,29], the hydrous nature and co-existence of +3 and +4 oxidation states can dramatically improve the electrochemical reactions [24,26].

### 3.3. Electrochemical Performance Analysis

We tested the electrochemical properties of the prepared samples by galvanostatic charge–discharge and linear cyclic voltammetry. The reaction equation in the electrochemical charge and discharge process can be expressed by the Equations (5) and (6) [24,27,28]:(5)$${\mathrm{MnO}}_{2}+{\mathrm{Na}}^{+}+e^{-}\leftarrow \to \mathrm{MnOONa}$$
(6)$${\mathrm{MnO}}_{2}+H^{+}+e^{-}\leftarrow \to \mathrm{MnOOH}$$

Due to the fact that the Mn^2+^ ion is water-soluble, it can be dissolved into the electrolyte [16], and this is in line with our XPS results.

As shown in Figure 5a, the electrodeposited solution with a concentration of 0.06 M has the highest specific capacitance, and this specific capacitance is 529.5, 322.5, 237.3 F g^−1^ at the current density of 1, 5, 10 A g^−1^, but when the concentration of precursor increases to 0.09 M, the specific capacitance decreased to 119, 62, 45.1 F g^−1^ at the current density of 1, 5, 10 A g^−1^. As the charge and discharge current increases, the specific capacitance of the sample decreases because the active material does not sufficiently react at a large current density. The specific capacitance calculated at different current densities can be used in different situations.

As can be seen from Figure 5b the charge and discharge curves of all samples are similar to isosceles triangles, and this shows that they have a highly reversible redox reaction, where between 1.0 V and 0.8 V there is a fast discharge area, which is mainly due to the decline in internal resistance (I.R drop), which is mainly due to the contact between the active material and the electrolyte and its own resistance change. Cyclic voltammetry is also one of the important means to detect the performance of electrode materials, wherein all samples have a pair of distinct redox peaks around 0.6–0.8 V. Results show that the 0.08 M sample has the highest area specific capacitance among the respective groups. As the concentration of the precursor increases, the rate of electrodeposition increases, and the thickness of the deposited sample also increases, which is why the area specific capacitance increases. 

However, when the concentration is increased to 0.09 M, the film thickness is maximized, the pores between the nanobars of manganese dioxide are reduced, and the utilization rate is the lowest, so the minimum area capacitance is exhibited (this is consistent with the previous SEM results). The specific capacitance of the MnO_2_ composites electrodes in 1 M Na_2_SO_4_ electrolyte reported in other literatures are shown in Table 1. Manganese dioxide electrode material with excellent properties can be obtained by electrodepositing a certain concentration of precursor on a foamed nickel substrate.

The chronoamperometry (CA) of samples was carried out to evaluate the stability of different MnO_2_ electrodes. Figure 5e shows the CA curve of MnO_2_ in 40 s at the given potential of 0.6 V. The current density value of all samples decreased rapidly in the early stage, and then gradually decreased slowly to a stable value. It was easily observed that the MnO_2_ electrode deposited into the 0.06 M precursor has higher electrochemical performance in terms of both the initial and steady states of the current density.

The cycling performance of the MnO_2_ (0.06 M) was evaluated by GCD measurements in 1 M Na_2_SO_4_ at a current density of 5 A g^−1^ for cycles up to 2000 (Figure 5f). The decrease in capacitance performance in the initial stage is mainly due to the dissolution of manganese dioxide in sodium sulfate solution, where during the testing phase we also observed that the sodium sulfate solution turned brown. Due to the improvement of the process, the adhesion strength of the deposited manganese dioxide has been improved, and the initial specific capacitance of 79.8% is still maintained after 2000 high current density (5 A g^−1^) cycles. 

Power density and energy density are also one of the important electrochemical properties of supercapacitors, as shown in Figure 6, the MnO_2_ deposited under the concentration of a 0.06 M precursor has a very high energy density (59.46 Wh Kg^−1^), and even at a power density of 3756 W Kg^−1^, it can reach an energy density of 24.9 Wh Kg^−1^. The energy density of other samples decreases with the increase of the precursor concentration, which is consistent with the GCD results.

EIS analysis is one of the effective methods to test the performance of electrode materials. Figure 6 reveals the Nyquist plots of the MnO_2_ electrode at a frequency range of 0.01–100,000 HZ (vs. Ag/AgCl) with 5 mV amplitude [33,34], the impedance data was analyzed by Zsimp Win V3.10 software.

As shown in Figure 7a, the prepared electrode exhibited typical capacitive behavior in 1 M Na_2_SO_4_ solution, In the low frequency region, the capacitor has a large resistance, while in the high frequency region, the resistance of the capacitor is very low (the impedance modulus is even lower than 1 Ω above 1 kHz). In the Nyquist plot, the high frequency region is semicircular, and the diameter of the semicircle represents charge transfer resistance (Rct), whereas the intercept on the real axis represents the solution resistance [11,35] (Rs, including the inherent resistance of the electrode, the contact resistance between the electrode and the electrode, etc.). In the equivalent circuit diagram, Cdl represents the electric double layer capacitor, Zw is Warburg resistance, which is caused by ion diffusion in the active material [36]. With the increase of the concentration of precursors, the Rs are (is) 1.183, 1.201, 1.299 and 1.312 Ω, and the Rct are 0.84, 0.93, 1.07 and 1.33 Ω, respectively. This result shows again that the 0.06 M sample has excellent supercapacitor performance.

## 4. Conclusions

In summary, manganese dioxide was directly electrodeposited on a foamed nickel substrate by changing the precursor concentration after ethanol immersion. An alcohol immersion process can improve the uniformity and electrochemical performance of manganese dioxide film, where all the obtained MnO_2_ material is a porous nanostrip, the diameter of the nanostrip is about 30–50 nm and the length of which is about 300–500 nm. XPS results show that Mn^3+^ and Mn^4+^ coexist in the prepared materials. The MnO_2_ electrode obtained in the 0.06 M precursor has a very high specific capacitance (529.5 F g^−1^ at the current density of 1 A g^−1^), low solution resistance (1.183 Ω) and charge transfer resistance (0.84 Ω); in addition, the energy density can even reach 59.4 Wh Kg^−1^ at the high power density of 3756 W Kg^−1^. By improving the process, the adhesion strength of manganese dioxide was increased, and the prepared manganese dioxide film has a capacity retention rate of 79.8% after being cycled 2000 times under a large current density. This work provided a promising direction for the electrodeposition preparation of high specific capacitance MnO_2_ supercapacitors.

## Figures and Tables

**Figure 1 materials-13-00181-f001:**
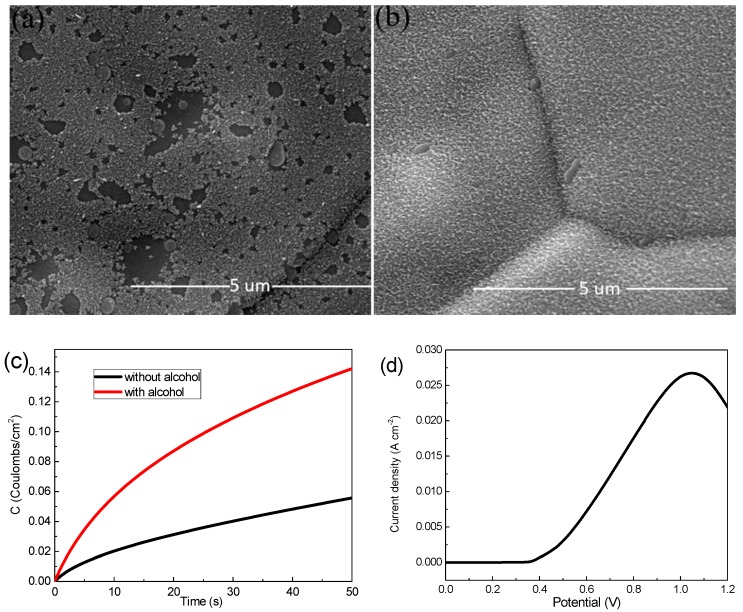
(**a**) Scanning electron microscope (SEM) images of samples without alcohol; (**b**) SEM images of samples with alcohol; (**c**) chronocoulometry technique curves with and without alcohol sample; (**d**) Linear sweep voltammetry (LSV) curve.

**Figure 2 materials-13-00181-f002:**
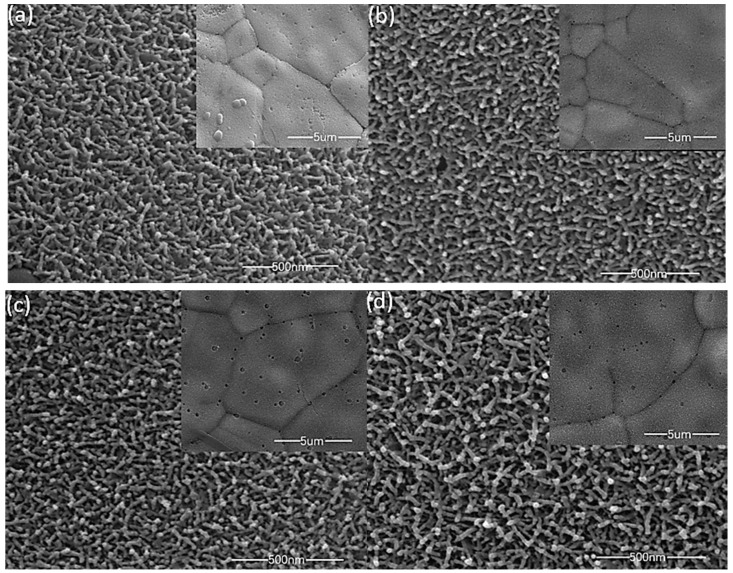
SEM images of different precursor concentrations; ((**a**), 0.06 M); ((**b**), 0.07M); ((**c**), 0.08M); ((**d**), 0.09 M).

**Figure 3 materials-13-00181-f003:**
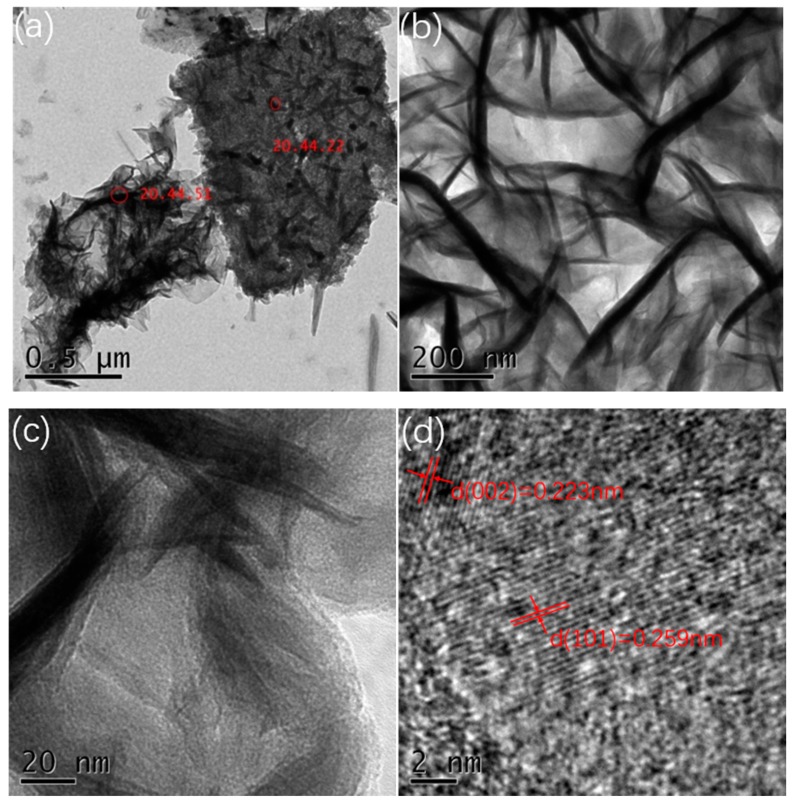
(**a**,**b**) TEM images at different magnifications; (**c**,**d**) HR-TEM images of the nanostrips from the 0.06 M sample.

**Figure 4 materials-13-00181-f004:**
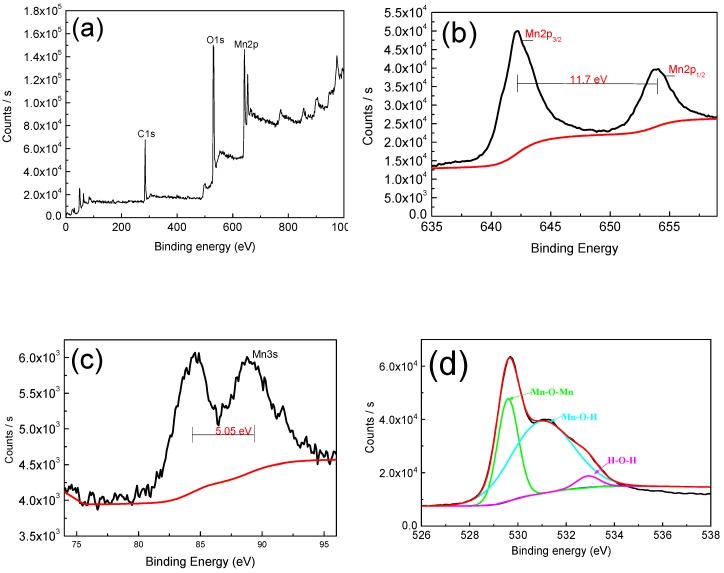
(**a**) X-ray photoelectron spectroscopy (XPS) spectra of MnO_2_ film; (**b**,**c**) Mn2p and Mn3s core level spectra; (**d**) O1s core level spectra.

**Figure 5 materials-13-00181-f005:**
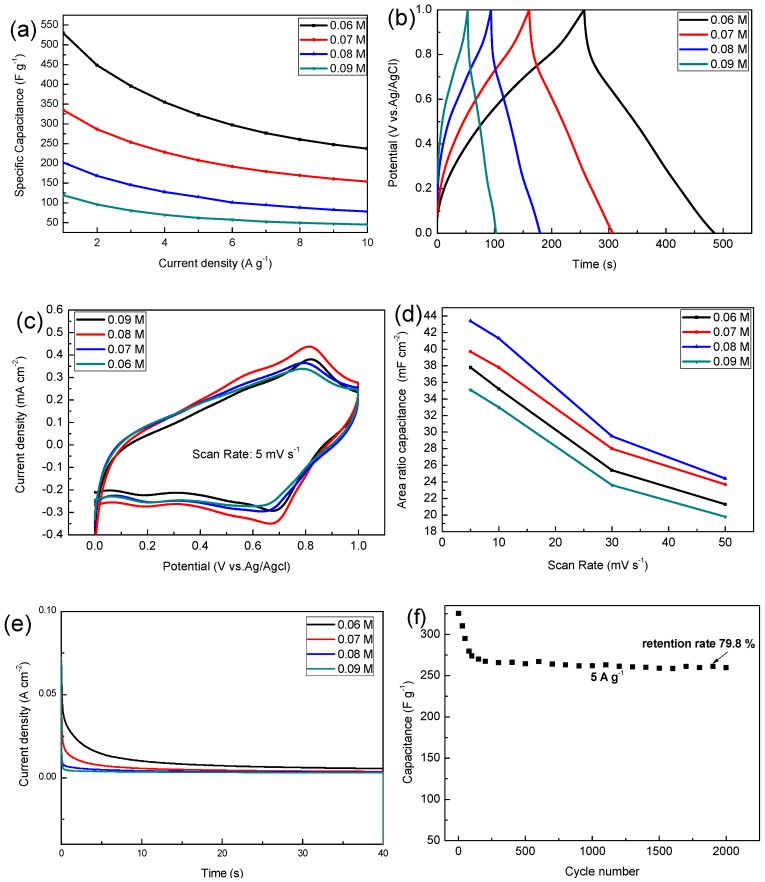
(**a**) The variations of specific capacitances of different current densities; (**b**) the charge and discharge time curve at a current density of 2 A g^−1^; (**c**) the cyclic voltammetry (CV) curves at a scan rate of 5 mV s^−1^; (**d**) area specific capacitance at different scan rates; (**e**) chronoamperometric response of different MnO_2_ electrodes; (**f**) cycling performance of MnO_2_ (0.06 M) at a current density of 5 A g^−1^.

**Figure 6 materials-13-00181-f006:**
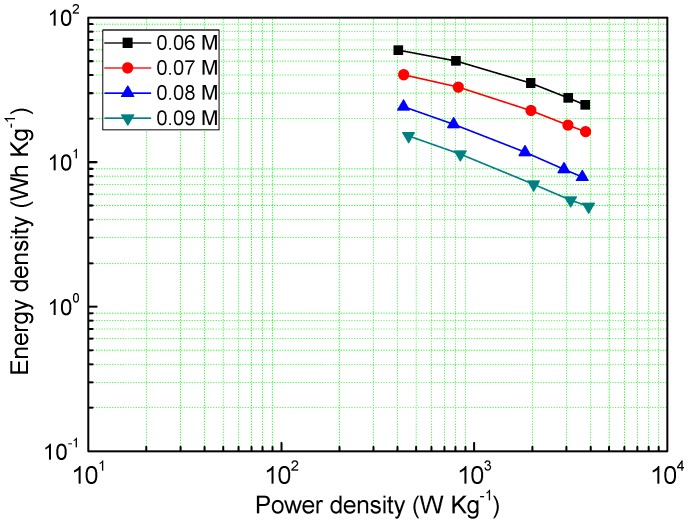
Energy density of different samples at different power densities.

**Figure 7 materials-13-00181-f007:**
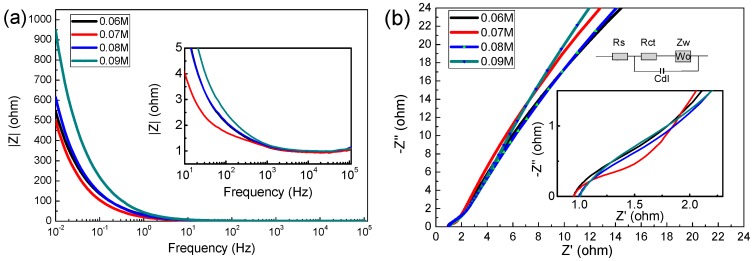
(**a**) Impedance modulus as a function of frequency; (**b**) electrochemical impedance spectra (Fitting circuit equivalent model are inset).

**Table 1 materials-13-00181-t001:** Specific capacitance compared with others.

Specific Capacitance	Current Density	Electrode	Reference
330 F g^−1^	1A g^−1^	MnO_2_@N-APC	[29]
262 F g^−1^	1A g^−1^	MnO_2_@CCNs	[30]
270 F g^−1^	1A g^−1^	MnO_2_	[31]
357 F g^−1^	1A g^−1^	MnO_2_	[32]
529.5 F g^−1^	1A g^−1^	MnO_2_	Our work

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
