# Peer review of "High Specific Capacitance of the Electrodeposited MnO2 on Porous Foam Nickel Soaked in Alcohol and its Dependence on Precursor Concentration"

_materials, 2020, doi:10.3390/ma13010181_

Round 1

Reviewer 1 Report

This work deals with an important topic, namely, the fabrication of supercapacitors with high power density and very good performance. To this purpose, the authors prepare thin films of manganese dioxide on porous nickel foam and study their characteristics. The results are interesting but their presentation is not clear enough, mostly because of the language problems. 

Comments:  the authors claim the dependence of the capacitance on the concentration of the precursor solution but this dependence is not clearly shown in the manuscript. There is not a "direct" deposition of MnO2, the material is formed from the precursor solution. Figure 1 is not clear. The role of the alcohol is not discussed. Also, it is not  discussed why the precursor solution having a concentration of 0.06M  gives such a performant film and not the other concentrations.

The manuscript will be more clear if the results will be explained more in detail.

Reviewer 2 Report

The authors reported a study concerning the electrochemical deposition of MnO2 films onto nickel foam substrates and its impact on the electrochemical performances in a 3-electrode cell configuration. More specifically, the corresponding coatings were obtained from a mixed aqueous solution of Mn(CH3COO)2 (precursor) and Na2SO4 tetrahydrate at different concentrations from 0.06 to 0.09 M, which were afterwards characterized at morphological (SEM, TEM), structural (XPS) and electrochemical (CV, GCD and EIS) level. From the energy storage point of view, the as-grown MnO2 films at 0.06M showed the best specific capacitance (SC) values. Thus, a SC value of 237.3 F g-1 was calculated at a current density of 10 Ag-1. After an exhaustive reading of the manuscript I have major concerns regarding the potential of this work, which are discussed as follow:

General comments:

a) One of the critical points of this study relies on the novelty and the state-of-the-art. In this context, numerous studies dealing with the impact of the electrodeposition conditions of MnO2 for supercapacitor applications have been already reported. I only cite some examples such as Variations in MnO2 electrodeposition for electrochemical capacitors, Electrochim. Acta 50, 2005, 4814, Morphology controlled MnO2 electrodeposited on carbon fiber paper for high-performance supercapacitors, Power Sources 351, 2017, 51, An effective electrodeposition mode for porous MnO2/Ni Foam composite for asymmetric supercapacitors, Materials 9, 2016, 246 among others. Honestly, I think the authors should revise the literature carefully and they should structure the introduction and the objectives of this work in a very different way.

b) The experimental section should be analyzed in-depth since important elements are missing.

b.1 How was the alcohol procedure obtained? Immersion time, temperature, thickness of the film, etc. Provide more details about this point.

b.2 For example, the author state that the deposition was conducted at 0.6V for 50s. How/Why did the authors choose that potential? Reference?

b.3 In the 2.3. characterization, the authors should explain how the samples were prepared for TEM measurements and provide more details about SEM (accelerating voltage)?

b.4 The quality of MnO2 were weighed with a high-precision electronic balance (Model, precision…?)

In overall, please provide a more detailed explanation of the experimental section.

c) Numerous misprints were detected throughout the text such as referenceelectrode (p. 2), remove etc in the abstract since all the employed techniques are already mentioned, Agcl, solution, In the (p. 7), nanstrips (p. 4), etc. Please, revise the text and English carefully.

d) In the introduction the authors state In this work, we systematically studied the effects of surface morphology, microstructure and electrochemical properties of manganese dioxide thin films deposited by different potentials in the potentiostatic method. However, the authors only employed the potential of 0.6V. Please, revise this point.

e) The authors should revise the presentation of the figures, Fig. 2 (h), areal capacitance instead of specific capacitance in Fig. 5d (y-axis), in Fig. 1 the authors show two SEM images but they are not clear and the scale bar is missing, Binding energy (eV is missing) in Fig. 4b, The order of the images in the caption of Fig. 2 is difficult to be understood (a,b, f, after c,g, later d, h, please order the figure for each concentration in a more logical manner, etc. Please, revise the figures again carefully.

f) The XPS in Fig. 4a shows peaks corresponding to Zn and Ca. Where do these elements come from?

Technical comments:

a) Specific capacitance was the only capacitive property studied in this work. In my opinion, other interesting properties should be analyzed such as electrochemical stability or energy density to characterize this material properly.

b) The authors reported specific capacitance values but these values were not compared to the state-of-the-art. How can the authors place this work in the literature in terms of performances? A summary table is necessary.

c) In the electrodeposition methods section the authors state that alcohol can adhere to the surface of the foamed nickel substrate, which is equivalent to covering a film on the surface, and then makes the thickness of MnO2 in the electrodeposition process more uniform. The authors provided two SEM images in Fig. 1 but they are not conclusive. Could the authors provide a cross-sectional view of both samples to confirm that statement? The authors should also explain the reason of the 'positive alcohol effect' regarding the MnO2 In this direction, could the authors compare the electrochemical performances with and without alcohol deposition?

d) Regarding the alcohol effect deposition the author state alcohol can adhere to the surface of the foamed nickel substrate, which is equivalent to covering a film on the surface. Could the authors provide more details in terms of characterisation about this film? What is the impact of this film on the electrochemical performances? Thickness effect?

e) Why do the authors employ a deposition potential of 0.6V? It should be explained.

f) Could the authors conduct BET measurements to evaluate the specific surface area? I think this information is highly recommended.

g) In my opinion, the IR drop in Fig. 5b is not negligible and the equation 1 should be adapted accordingly

h) What is the thickness for the respective systems 0.06, 0.07, 0.08 and 0.09M?

j) Could the authors display the CV for the oxidation potential at 0.6V and the corresponding chronoamperometry at 50s.?

Based on my previous arguments I believe this work needs still major, major revisions to be considered for publication in the journal Materials.

Round 2

Reviewer 1 Report

This revised manuscript has been much improved compared to the first version. The authors reponded most of the comments and questions.

Author Response

Thank you very much for your recognition, review and suggestions for our work. We have revised some grammar and spellings and other issues of the manuscript again and marked them in red.

Reviewer 2 Report

The authors revised the manuscript according to my comments carefully. I have enormously appreciated the work and effort conducted in this revision, which provided more clear understanding and improvement of this study. However, I consider there are several points, after the author’s responses, to be analyzed properly. In this context, I have some questions as follow:

The authors reported the GCD and cycling stability of both alcohol and without alcohol deposition methods. An enormous difference regarding the specific capacitance between both systems is clearly evidenced (approximately 460 and 160 Fg-1), demonstrating the effect of the surface by using alcohol. Surprisingly, after only 20 cycles the SC value was reduced up to 350 F/g, corresponding to a loss of 23%!!, whereas in the particular case of 'without alcohol' a slight decay was observed. From this point of view, it seems the cycling stability of the alcohol-based system is extremely low. Could the authors analyze this property? How is the cycling stability after thousands of GCD cycles? This property is critical for supercapacitor applications and whether the aforementioned tendency is confirmed, this is not a material for supercapacitor applications.

Concerning the alcohol factor, I think this point is omitted in this study and its impact is enormous dealing with the electrochemical performances reported by the authors. Undoubtedly, I agree the influence of the precursor concentration in this work is important, but I believe the authors can not neglect this phenomenon, which is crucial and so decisive as the precursor concentration. In my opinion, this point should be highlighted in conclusions, abstract, modification of the title, etc, as an example. Please, highlight the impact of the MnO2 deposition onto alcohol-treated nickel foam substrates

As I mentioned in my previous report the SEM images in Fig. 1 are blurry and they are not conclusive to state that in the case of alcohol the MnO2 coating is more uniform. Please, revise the Figure 1 accordingly. In this context, the authors should provide a cross-sectional view to confirm this statement. Regarding this last point I disagree with the authors, it is possible to measure the thickness by means of SEM/FIB techniques even for porous nickel foam substrates. Please, check this point carefully.

Concerning the electrochemical performances reported in the revised document I have several questions to evaluate the effect of alcohol. Could the authors compare both with and without alcohol at the same charge by using the chronocoulometry technique?. It can happen that at the same time and potential the total charge is different leading to different mass variations. Thus, several factors are playing at the same time restricting the effect of alcohol.

The LSV graph should be included in the main text.

Honestly, I do not think the introduction reflects the reality of this research topic. The effect of different precursors, electrolytes and electrochemical conditions among others have been widely investigated regarding the electrochemical deposition of MnO2. The introduction is very brief compared to the vast literature in this topic. Probably, a more exhaustive analysis is highly recommended.

Please revise the format of the new references according to the journal.

Numerous misprints were still detected in the text. Please, revise again this part

English should also be revised.

Round 3

Reviewer 2 Report

The authors revised the manuscript according to my comments carefully. All the points were exhaustively examined, and therefore I recommend this article for publication in the journal Materials. However, the quality of some new added figures (e.g. axis of Fig 1c and 1d) should be revised as well as English.

Author Response

Thank you very much for your help and suggestions, we have done our best to correct the picture information and grammar issue of manuscript,and increased the discussion of possible mechanisms of alcohol and data on circulation performance, all changed have marked in yellow.